# Evaluating an Augmented Reality Interface for Drone Search Tasks

### David Kortenkamp
korten@traclabs.com
TRACLabs Inc.
1331 Gemini Street, Suite 100
Houston, TX, USA 77058

### Debra Schreckenghost
schreck@traclabs.com
TRACLabs Inc.
1331 Gemini Street, Suite 100
Houston, TX, USA 77058

### Gavin Love
gavin@diamondagetechnology.com
Diamond Age Technology
Houston, TX, USA 77058

### Patrick Hagan
Patrick.Hagan@outlook.com
Houston Fire Department
Houston, TX, USA 77058

## ABSTRACT

Drone operations are an intriguing application for augmented reality interfaces. These operations involve controlling a six-degree-of-freedom drone to accomplish tasks while also maintaining situational awareness with respect to the surrounding environment. We have developed a prototype augmented reality interface for drone search-and-rescue operations. This interface runs on a Microsoft HoloLens 2 headset and displays drone telemetry, position, targets, and tasks. We then performed an evaluation of this interface using seven experienced drone pilots. This evaluation included user satisfaction and workload. We present an overview of our augmented reality interface and the results of our evaluation.

**ACM Reference format:**
David Kortenkamp, Debra Schreckenghost, Gavin Love, and Patrick Hagan. 2024. Evaluating an Augmented Reality Interface for Drone Search Tasks. In *Proceedings of VAM HRI, Boulder, CO, March 11, 2024,* 7 pages.
https://doi.org/10.1145/nnnnnnn.nnnnnnn

## 1 INTRODUCTION

Drone operations require a drone pilot to both look down at a tablet screen to see drone telemetry while also keeping track of the location of the drone in the sky. Sometimes, this requires a second person (called a visual observer) to assist the pilot. As drone operations become more common in first responder scenarios (e.g., law enforcement and fire) an augmented reality interface for use during drone search tasks may be useful. We developed a prototype of such an interface and performed a user study with first responders and other drone operators. The augmented reality interface complements the existing drone interfaces by providing additional, context-sensitive information. Augmented reality is used in four ways. First augmented reality displays drone telemetry information (e.g., altitude, speed, battery level, etc.) in a display that is overlaid

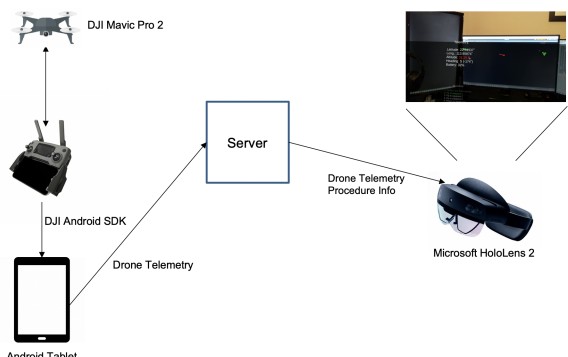

**Figure 1: Software architecture for the drone user study.**

on the user's field of view. Second, augmented reality shows the current location of the drone and the target with respect to the user by highlighting each on the display. Third, augmented reality shows a list of tasks to be completed and the completion status of each task. Fourth, the augmented reality shows a live, overhead map of the location of the drone, the targets, and the operator. In this paper, we describe both the prototype augmented reality interface and the results of our user study with first responder and civilian drone pilots.

## 2 AUGMENTED REALITY INTERFACE

We developed a prototype augmented reality interface that connected to live drone telemetry and displayed visual cues to a drone operator. In this section, we discuss both the hardware and software infrastructure and the augmented reality interface.

### 2.1 Display device and software infrastructure

The Microsoft HoloLens 2 was used as the display device for the augmented reality (AR) system described in this paper. The HoloLens 2 has a see-through plastic lens with a 2K resolution display built-in. The HoloLens 2 has a horizontal field of view of 43 degrees and a vertical field of view of 29 degrees. The HoloLens 2 performs head and eye tracking and also has a video and depth camera. Tracking of left and right hands is performed by on-board cameras. The

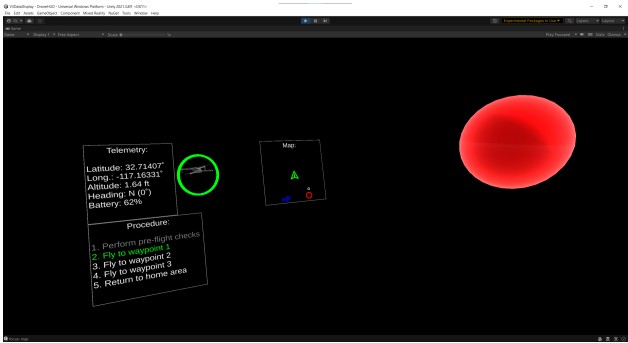

**Figure 2: Screenshot of the drone 3D AR interface.**

HoloLens 2 runs a version of Windows on-board the device. We developed our AR application in Unity[1] and then loaded that application onto the HoloLens 2. The AR application connected to live drone telemetry. Telemetry was pulled from the drone using the DJI Android Software Development Kit (SDK) and a Python script running on an Android tablet. The Python script posted the drone telemetry to a REST server as a JSON structure. The Unity app used a REST API to get the drone telemetry from the server and posted these data to a live AR display. The telemetry was updated once a second. Figure 1 shows the overall software architecture.

## 2.2 Augmented reality prototype

The augmented reality interface had a number of elements and these are shown in Figure 2. In that figure, everything in black is transparent and would show the real world when displayed in the HoloLens. The upper left element contains live telemetry from the drone. The lower left element shows the tasks that need to be performed. The tasks are automatically completed based on drone telemetry. The middle element shows a green circle that indicates the drone's current location based on drone telemetry. The upper right element shows a 2D, overhead map with the drone (green triangle), user (blue icon), and target (red circle). The green triangle is oriented with respect to the drone heading. The red circle on the right of the display is actually a sphere and represents the target location. The red sphere is fixed to a pre-defined location in the display. The green circle and the red sphere do not appear if the drone or target are not in the user's field of view. In these cases, a green and red arrow show the direction to the drone and the target respectively. After the drone reaches a target, a new target location (red sphere) appears and the task list is updated. Figure 3 shows a different view of the interface. This shows the green and red arrows and also the ability to highlight telemetry that might be of interest to the user. Note that the 3D drone model that is shown in these screenshots is for clarity and was not shown during the user study. It could be shown optionally for debugging the system and when a real drone was not connected.

There was an additional step necessary for the system to be operational. The initial location of the drone relative to the HoloLens headset needed to be set. This was done prior to the run by using a game controller to line up a 3D model of the drone on top of the

---

[1]unity.com

**Figure 3: A second screenshot of the drone 3D AR interface.**

actual drone while the drone was on the ground in front of the user. Once the 3D model was aligned in all 6 axes with the actual drone the system was calibrated. The user could then move around and also fly the drone around. Recalibration was not necessary until the system was restarted. For the user trials, in the interest of time, calibration was done by TRACLabs staff before handing the headset to the study participant.

## 3 RELATED WORK

There have been several prototypes developed for using an AR interface for drone inspection, including bridges [11] and buildings [7]. The former implemented only a virtual reality simulation of the AR interface and conducted a user study in that VR environment. The latter focused not on actually flying the drone, but on using AR to integrate drone imagery with building schematics. A similar project focused on using an AR interface to define paths for a drone to follow while performing an inspection [1]. Again, this research did not address the real-time drone pilot interface. Another research prototype integrated both HoloLens gesture recognition for drone control and AR presentation of drone video and telemetry[6]. Their user study focused exclusively on the gesture-based drone control. They do provide some interesting future directions for work of this type, including the integration of live drone imagery and the use of image recognition to track drone location instead of relying on drone IMU and GPS sensors. This paper also noted the need for manual calibration between the drone location and the headset location, which is a finding we observed as well. Image recognitive for drone tracking could remove this manual calibration step. Recent work in AR for robotics in search and rescue applications has focused on recreating the 3D world model being provided by the robots and drones [12], but this work does not address the experience of the drone pilot. Work on communicating aerial robot intents focused on three different displays: 1) planned flight paths relative to current drone location; 2) an arrow in 3D space showing the future drone location and direction; and 3) a virtual "eye" that stares at the drone's destination [13]. This work included a user

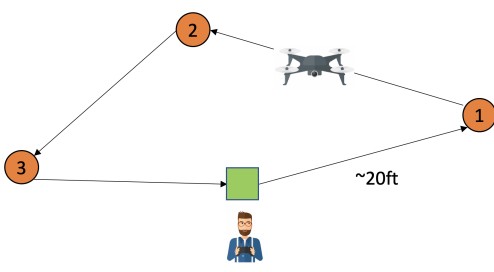

Figure 4: Scenario for the drone user study.

study with the three displays focused on task performance and inefficiencies. Other work focuses on displaying the live drone camera feed in an AR interface [4]. We did not display drone camera feeds in our display, but this is an interesting future direction. A commercial company, Anarky Labs, is selling a HoloLens drone interface[2]. However, there is not much information available nor any documented user studies.

## 4 USER STUDY

After developing the augmented reality interface and informally testing it using licensed drone pilots, we conducted a user study to evaluate its effectiveness. We submitted our study protocol to our Institutional Review Board (IRB), which declared our study exempt from IRB oversight. The funding agency for this study, the National Institute of Standards and Technology (NIST), also reviewed the study protocol with respect to its human research guidelines and provided its approval. This section describes the user study protocol, participants, and data measures.

### 4.1 Scenario

The drone scenario was a search task. The participants were asked to go to pre-defined target locations that were identified both in the AR interface and with cones on the ground (see Figure 4). The goal was not to test the pilots on their accuracy or speed in performing the search task, but to instead to measure the interface's usability and workload, elicit the participant's feedback about the interface, and determine how it helped the participant perform the search task. At times, the participants were asked to look away from the drone and to re-acquire its location visually. For the study course, the drone was always within 25 feet of the participant, so the time to re-acquire after looking away was small both with and without the AR interface. Participants did feel that if the drone course were much larger (larger than we could accommodate) then the re-acquisition timing difference might be noticeable.

### 4.2 Methodology

Participants each filled out a pre-session survey that captured basic demographic information and the participant's familiarity with drones, augmented reality, computers, and computer games. Each participant started with a short training period on flying the drone

[2]https://anarkylabs.com/airhud-for-hololens-2/

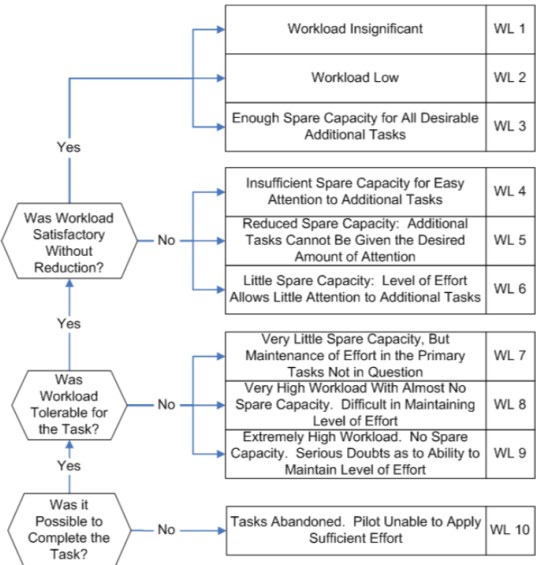

Figure 5: Flowchart for calculating workload (taken from [10]).

to understand the existing drone piloting interface. We had different levels of experience in drone operations and with the drone we are using, so the training session was short for pilots experienced with our drone and longer for novice pilots. The participant then flew a short drone "course" with a sequence of targets. Participants were asked to hover at each target and verbally call it out. The course was flown using the augmented reality interface described in the previous section. The participant was distracted once and told to reacquire the drone visually. After flying the course, participants were allowed to fly the drone freely to investigate different aspects of the augmented reality display. Surveys of cognitive workload, system usability, and a smart glass user satisfaction were administered after the run. Finally, an overall assessment questionnaire and a free-form interview was conducted.

### 4.3 Research hypothesis

Our research hypothesis for this study was that the AR interface is a usable display with acceptable workload for flying a drone for search tasks.

### 4.4 Data measures

Quantitative measures were collected to provide insight into the usability and workload when using the augmented reality interface and to determine whether there are potential adverse effects of the interface. Subjective measures computed for this study include workload measured using the Bedford scale (see Figure 5 and [9]), and user satisfaction and usability of AR interface, measured using a modification of the System Usability Scale (SUS) [3] and Smart Glass User Satisfaction (SGUS) [5] questionnaires. Participants were also asked to volunteer feedback on their session that was collected for anecdotal analysis

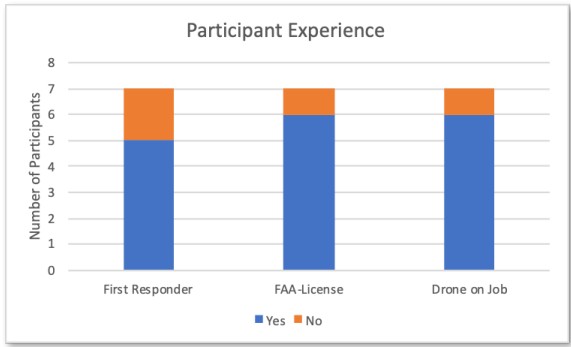

Figure 6: Participant experience as first responder and in flying drones.

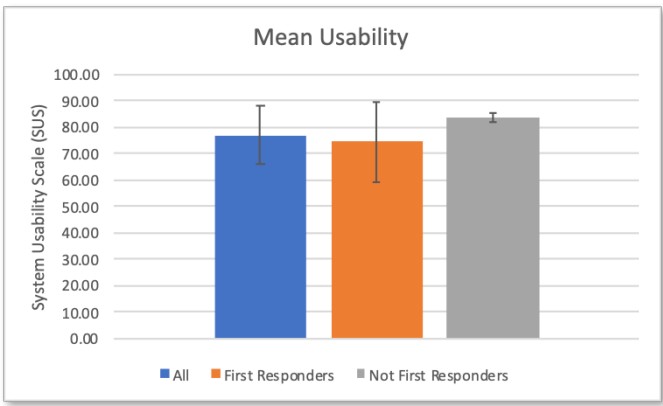

Figure 7: Mean usability using SUS for drone study.

### 4.5 Participants

The study had a total of seven participants, six male and one female. All participants were drone pilots. Five participants were first responders, and six were FAA-licensed drone operators. Six participants had used a drone on the job. Thus, all participants are very representative of potential users of an augmented reality interface for first responders, as summarized in Figure 6.

Five participants had between 100 and 200 hours of drone flight time. One participant was very experienced, with 1500 hours. And one participant was less experienced, with 20 hours of drone flight time. Six participants had some prior familiarity with the DJI Mavic Pro2 drone, the drone used for this study. Four were very familiar and two were somewhat familiar.

### 5 FINDINGS

In this section we report findings with descriptive statistics from this study.

### 5.1 Usability

The primary measure used to assess the usability of the AR interface is the System Usability Scale (SUS) [3]. SUS poses five positive and five negative statements. The user rates the statement from

| SGUS Statements | All | First Responders | Not First Responders |
|---|---|---|---|
| With AR-glasses I could access information at the most appropriate place and moment | 4.1 | 4.2 | 4.0 |
| AR-glasses allowed a natural way to interact with information displayed | 5.1 | 4.6 | 6.5 |
| AR-glasses provided me with the most suitable amount of information | 5.3 | 5.2 | 5.3 |
| While using AR-glasses, I was aware of the phase of the task at all times during the execution of the task | 5.7 | 5.6 | 6.0 |
| While using AR-glasses, I was able to pay attention to the essential aspects of the task all the time | 5.7 | 5.6 | 6.0 |
| Content displayed on the AR-glasses made sense in the context I used it in | 5.9 | 6.0 | 5.5 |
| The instructions given by AR-glasses helped me to accomplish the task | 5.9 | 5.8 | 6.0 |
| The interaction with content on AR-glasses captivated my attention in a positive way | 6.1 | 6.4 | 5.5 |
| I understood what is expected from me in each phase of the task with the help of AR-glasses | 6.3 | 6.2 | 6.5 |
| Performing the task with the help of AR-glasses was natural to me | 6.4 | 6.4 | 6.5 |
| I had a good conception of what is real and what is augmented when using AR-glasses | 7.0 | 7.0 | 7.0 |

Figure 8: Mean user satisfaction using SGUS for drone study.

1 (Strongly Disagree) to 5 (Strongly Agree). Responses are then combined in a weighted equation to compute a usability number between 0 and 100. For SUS, larger numbers indicated better usability. The mean SUS measure for the AR interface across all participants is 77.1 +/- 11.2. SUS for the five first responders is 74.5 +/- 15.1. The two non-first-responders found the system somewhat more usable, at 83.75 +/- 1.7. Figure 7 summarizes the SUS measures. A SUS score of 70 or above is generally considered acceptable for a user interface [2].

A second questionnaire measuring usability was administered after the session using the AR interface. The Smart Glasses User Satisfaction (SGUS) [5] measures user satisfaction with the information displayed in smart glasses and AR headsets. SGUS poses a series of statements, and asks the user to rate each one from 1 (Strongly Disagree) to 7 (Strongly Agree). All statements are phrased positively, so a higher rating indicates a more favorable response. Figure 8 lists the SGUS questions, with the mean response from participants. Rows are sorted from the least favorable response to the most favorable response in the "All" column.

Overall, 10 of 11 statements received a favorable response. The statement with the lowest mean response of 4.1 was "With AR-glasses I could access information at the most appropriate place and moment." The lower response to this statement was likely influenced by unreliable access to telemetry and inaccuracy in drone location information experienced by some users.

Statements with weak agreement (on average 5.1 and 5.3 respectively) include "AR-glasses allowed a natural way to interact with information displayed" and "AR-glasses provided me with the most suitable amount of information." This indicates a need to improve both what information is displayed and how users interact with it. First responders, however, rated the first statement almost 2 points lower than non-first-responders.

First responders rated this statement nearly 1 point higher than non-first-responders – "The interaction with content on AR-glasses captivated my attention in a positive way." This may indicate that

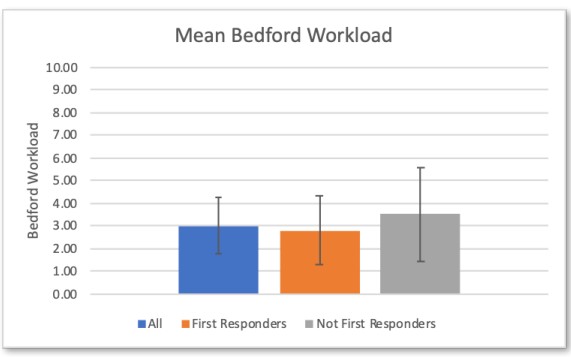

**Figure 9: Mean Bedford workload for drone study.**

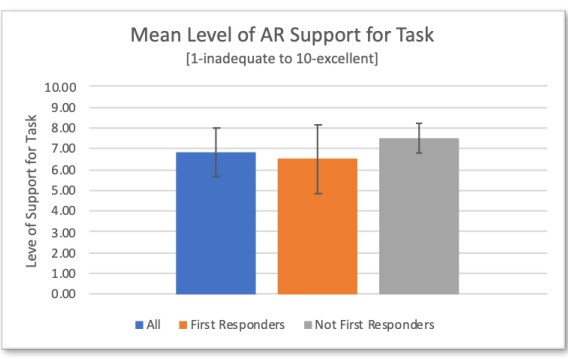

**Figure 10: Mean level of augmented reality support for the task in drone study.**

first responders were less familiar with augmented reality technology than non-first-responders.

The statement with the highest mean response of 7 was "I had a good conception of what is real and what is augmented when using AR-glasses."

These values compare favorably with those found in the original SGUS paper [5]. In that paper, five of the questions had weak agreement (4.8 to 5.7) whereas two of the questions had strong agreement (5.9 or 6.0).

### 5.2 Workload

Workload was measured using Bedford Workload scale [9] (see also Figure 5). Five of seven participants experienced low workload, at 3 or below. But two participants experienced higher workloads of 5 or 6. Both users with higher workload made repeated attempts to move the drone close enough to the target for the AR interface to recognize the target was reached. They were unable to reach the target due to imprecision in the GPS-based drone location and to target proximity to an obstacle (i.e., a nearby tree trunk). Additionally, the drone telemetry server went down multiple times during one user session, requiring pausing and restarting it during the run. This user rated the workload as higher. Figure 9 summarizes the workload measures. A NASA technical brief states that a workload of 3 or less on the Bedford scale is the target for interfaces for nominal tasks [8]

### 5.3 Task support

After flying the drone using the augmented reality display, participants were asked to "Rate how well the Augmented Reality aid supported you in performing the task." They used a scale from 1 (inadequate support) to 10 (excellent support). On average for six participants, the aid was rated at 6.8. One first responder participant did not complete the post session survey. This rating was 6.5 +/- 1.63 for the remaining four first responders, and 7.5 +/- .69 for the two non-first-responders. Figure 10 summarizes this task support measure.

Users also were asked whether they would use the augmented reality display to fly the drone. Five of six participants indicated they would want to use the heads-up display for drone flying.

When asked about the usefulness of a specific feature of this display, 5 of 6 participants would use the telemetry display, the

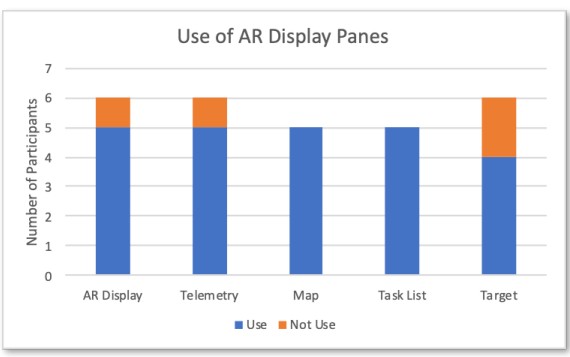

**Figure 11: Number of Participants that would use each of AR display panes.**

map, and the task list. 4 of 6 would use the target sphere. Figure 11 summarizes these results. Two of six participants did not find the target feature useful.

### 6 DISCUSSION

This study has the hypothesis that positive user satisfaction and usability will be reported when using the drone AR display to fly the drone. Based on the findings reported above, we conclude our AR display was considered usable for drone flying. The overall SUS value was 77.1 +/- 11.2. The SGUS measures indicated that 10 of 11 statements received a favorable response, and one statement was considered neutral with a value of 4.1 on a 7-point scale. Participants also rated how well they felt the drone AR display supported the task of flying the drone to specified target. On average for six participants, the aid was rated at 6.8 +/- 1.2, on a scale from 1 (inadequate support) to 10 (excellent support). Bedford workload was low, with a mean value of 3 +/- 1.25. Higher workload corresponded to session where there were problems with the accuracy and reliability of the AR technology. The mean level of AR support for the task of flying the drone was rates above average at nearly 6.8 (see Figure 10). Additionally. a majority of the participants reported they would use all panes in the AR display (see Figure 11 for details). Taken together, these findings indicate a positive user satisfaction with this display.

We summarize other important feedback from participants about the use of augmented reality for drone operations in the remainder of this section.

## 6.1 AR to Find the Drone

The AR user interface has the potential to assist the operator in locating the drone after distraction, comparable to the assistance provided by the Visual Observer member of the drone team. For the site of the drone session, the drone was never far enough away to become hard to find. Additionally, inaccuracy in the drone location meant the overlay locating the drone was not always over the drone, making it hard to get accurate drone location. We did get positive feedback from a number of participants (6 of 7 participants used the green circle to reacquire the drone) indicating that such a feature would be very valuable to them. Some participants said they don't have virtual observer, so such a feature would be very helpful. Participants indicated this would prove particularly useful as the drone moves further away or when the drone is flying in areas with visual clutter. One participant suggested that the green circle also indicate which way the drone is pointed.

## 6.2 AR to Find the Target Location

Currently the Drone AR app highlight planned waypoint locations using a red sphere. When the drone flies into the sphere, the way-point is considered 'reached' and the red sphere is moved to a new waypoint location. When this worked, participants liked it. We found that drone location based on GPS data was not sufficiently accurate to reliably detect when the drone had reached a target. Additionally, obstacle avoidance on the drone prevented getting too close to targets that were near terrain features (like trees). As a result, some waypoints were never shown to the drone operator. The app should be modified to allow users to mark a waypoint as 'reached' when automated detection is not possible. One participant suggested an audible tone be used to indicate when a target had been recognized. Techniques should also be investigated to improve the accuracy of the drone's location.

## 6.3 AR Display of Drone Telemetry

Almost all participants felt a display of drone telemetry to be useful while flying the drone. Which telemetry data should be displayed, however, varied by use of the drone, suggesting the need for some configurability for each use. For example, when using the camera for surveillance, it would be useful to have camera gimbal angle. Other suggested data include drone direction, air speed, pressure, wind direction, wind speed, and GPS signal strength. Most users felt latitude and longitude were not needed. There was some discussion of annotating telemetry to indicate kinematic limitations or possible threats, such as getting too close to the tree tops. Some first responders felt data values would be more useful if presented graphically instead of textually. Text displays that are frequently accessed can take too much time to read, and thus be distracting. There were some exceptions, however, such as text messages tied to dispatch during search-and-rescue (SAR), to update status on a missing person.

## 6.4 AR Display of Task List

Most users felt the Task List could be useful, although some felt it would be more useful in drone training than in operations. For some uses, there is only one task at a time, so a task list isn't needed.

## 6.5 Layout of AR Display

Two users indicated that the current layout of virtual forms required too much eye and head motion. They felt they were always in motion to scan them. This was considered distracting as well as effortful. A third user felt the telemetry pane was moving in and out of their field of view (FOV) with head motion, and would like it to remain in their FOV. This participant felt the Map should be moved to lower right so the telemetry would be more prominent. Another participant stated they would like to change the position of virtual forms based on preference or even task, because sometimes they were in the way. Additional work on placement of virtual forms is needed to address some of these issues.

## 6.6 AR Used Outside

Providing for configurability of the overlays was suggested by most participants. The visibility of text (currently shown in white font) varied quite a bit, based on changes in lighting due to time of day or whether the day was sunny or cloudy. Suggestions included changing text and graphic colors and transparency as needed to provide best contrast with environment and lighting conditions in a particular situation. One user suggested the use of color themes that can be easily changed while flying (possibly using voice command).

## 7 CONCLUSIONS

Augmented reality is a fairly novel area of research for drone piloting. A recent survey of AR applications for robots, identifies very few AR interfaces for drone piloting [14]. Most drone controls are either tablet-based or first-person, immersive virtual systems. In both cases, situational awareness is compromised for the pilot. In general, our study showed that AR systems have potential to provide drone pilots with information that they would otherwise need to get from looking down at a tablet and taking their eyes off of the drone. The AR system also has the potential to provide information that a tablet-based display cannot – primarily to locate the drone in the pilot's field-of-view. This latter capability was universally mentioned as the most intriguing capability, especially if piloting a drone over long distances or beyond visual line-of-sight (BVLOS). The AR interface also has potential for use by a visual observer who is not flying a drone. In some departments with BVLOS operations, a visual observer is still used (often on roof-tops) to keep an eye on the drone. They could benefit from a heads-up display of this type. While indicating the location of the drone in the pilot's field-of-view was well received, the reliance on GPS was a problem. GPS can have errors of several meters in certain conditions (under trees, near large buildings, etc.) and sometimes in our trials the indicated location of the drone in the AR system was not aligned with the actual location. This happened more often when the drone was close to the pilot and close to the ground. A vision-based drone location ability would help with this (maybe with a QR code of some type), but this would not work if the drone is hidden behind an obstacle. The requirement to manually calibrate the headset to the drone

was also cumbersome. This could also potentially be helped with a QR code on the drone. Finally, the equipment we used (Microsoft HoloLens 2) was not well-suited for use outdoors in bright sunlight. The overlaid cues were often washed out when pilots look up into the sky (as they often do when flying a drone). Future work will focus on addressing the concerns raised by study participants, adding a live drone camera feed, and displaying information about other drones that might be in the workspace.

## 8 ACKNOWLEDGMENTS

This work was sponsored by the National Institute of Standards and Technology (NIST) through their Public Safety Communications Research (PSCR) program under contract number 1333ND20PNB670731. Special thanks to Scott Ledgerwood of NIST PSCR for input into the research questions examined in this paper. Thanks to Tod Milam of TRACLabs for his work on the software to export telemetry data from the drone to the REST server.

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
