# OpenReview forum: "Evaluating an Augmented Reality Interface for Drone Search Tasks"
_humanrobotinteraction.org/HRI/2024/Workshop/VAM-HRI — VAM-HRI 2024 Oral_

### Official Review · Reviewer_VUVs · 2024-02-03
**Review 1: Accept**

**Rating:** 6
**Confidence:** 5

**Review:**

Summary:
This paper describes an augmented reality interface for supporting the tele-operation of a drone, geared towards first responders. The goal is to reduce the workload of controlling a drone by providing various visual design elements to support users. Results suggest that augmented reality can be utilized to support drone operators, with multiple lessons learned throughout the initial user study.

Strengths:
- This paper presents an AR drone interface, an understudied area of VAM-HRI.
- This paper also conducts a user study with the intended user population (first responders), gaining valuable insight from their feedback.
- Numerous lessons learned are presented in the discussion and provides valuable insight for future AR drone interfaces.

Weaknesses:
- It is unclear how the user study environment was set up. It would be helpful if Figure 4 was an actual picture of the environment along with the search targets.
- The visual design elements seemed to support 2D interfaces better than 3D. For example, rather than a 2D map of the environment, perhaps it would have been more helpful if it was displayed in a world-in-miniature style [1]. That way, operators may understand the 3D position of the drone. While the paper describes what was implemented, it could be strengthened by describing why certain UI elements were included based on existing literature.
- It is stated that, “ The goal was not to test the pilots on their accuracy or speed in performing the search task, but to elicit their feedback about the interface and how it helped them with the task.” Therefore, it seems like the user study should have included a baseline to compare against. It could be discussed in future work whether direct comparisons between industry standard interfaces with the AR interface in a within-subjects study might provide deeper insight into the benefits and drawbacks of the AR interface.

Other relevant citations might include: [2-4]

[1] M. Walker et al., "A Mixed Reality Supervision and Telepresence Interface for Outdoor Field Robotics," 2021 IEEE/RSJ International Conference on Intelligent Robots and Systems (IROS)
[2] A. Angelopoulos et al., "Drone Brush: Mixed Reality Drone Path Planning," 2022 17th ACM/IEEE International Conference on Human-Robot Interaction (HRI)
[3] Michael Walker, Hooman Hedayati, Jennifer Lee, and Daniel Szafir. 2018. Communicating Robot Motion Intent with Augmented Reality. In Proceedings of the 2018 ACM/IEEE International Conference on Human-Robot Interaction (HRI '18)
[4] Hooman Hedayati, Michael Walker, and Daniel Szafir. 2018. Improving Collocated Robot Teleoperation with Augmented Reality. In Proceedings of the 2018 ACM/IEEE International Conference on Human-Robot Interaction (HRI '18)

---

### Official Review · Reviewer_pXBY · 2024-02-05
**Review 2**

**Rating:** 6
**Confidence:** 5

**Review:**

Summary: This paper presents a novel augmented reality (AR) interface to act as a heads-up display for aerial drone pilots, allowing operators to simultaneously attend to streams of telemetry while still seeing the environment, thus maintaining improved situational awareness. The authors present an exploratory evaluation, testing the interface in a navigation task with experienced first responder drone pilots, and reporting survey measures of usability and workload, alongside design feedback from participants.

Strengths:
- The novel AR interface is well-tailored to a specific problem (needing to see streams of telemetry while also maintaining visual tracking of a drone's location).
- The paper's use of first responders and experienced drone pilots as participants provides invaluable feedback and increases the validity of the findings. It's relatively rare to see user studies properly recruit participants with the degree of expertise that active users in the target domain would have.
- The feedback from the study and resultant discussion is useful knowledge for informing the design of AR-based interfaces for drone teleoperation.

Weaknesses/Suggestions for Improvement:
- The paper could use some additional grounding/exploration of the state of the art for both drone operation interfaces, and the use of AR for interacting with aerial robots. For a paper of this length, it's worth expanding the related work section beyond its current length of one paragraph, citing additional research to provide better context to readers.
- There are no images of the interface being used in-situ, which makes Figures 2 and 3 less helpful for deciphering how the interface is used in practice. Would recommend including a figure with a screenshot of the HoloLens interface, overlaid with the actual testing environment.
- The user study would be much stronger if it contained comparison to one or more alternative interfaces, especially more traditional drone teleoperation interfaces. In a vacuum, it is hard to quantitatively determine what improvements the novel interface affords.
- Relatedly, the research hypothesis is underdefined. A hypothesis should be a testable statement. Therefore, "usable display" and "acceptable workload" should be defined upfront and grounded to their respective survey scales. Ideally, these would be comparative hypotheses measuring usability/workload across experimental conditions, but given the single condition, benchmarking against common thresholds in the various pre-validated surveys is likely the best option.

Recommendation: This paper would be a good addition to VAM-HRI 2024, and I recommend it be accepted.

---

### Decision · Program_Chairs · 2024-02-05

Accept (Oral)